# Optimized Metabotype Definition Based on a Limited Number of Standard Clinical Parameters in the Population-Based KORA Study

**DOI:** 10.3390/life12101460

**Published:** 2022-09-20

**Authors:** Chetana Dahal, Nina Wawro, Christa Meisinger, Taylor A. Breuninger, Barbara Thorand, Wolfgang Rathmann, Wolfgang Koenig, Hans Hauner, Annette Peters, Jakob Linseisen

**Affiliations:** 1Independent Research Group Clinical Epidemiology, Helmholtz Zentrum München, German Research Center for Environmental Health (GmbH), Ingolstädter Landstr. 1, 85764 Neuherberg, Germany; 2Epidemiology, Faculty of Medicine, University Hospital Augsburg, University of Augsburg, Stenglinstraße 2, 86156 Augsburg, Germany; 3Institute of Epidemiology, Helmholtz Zentrum München, German Research Center for Environmental Health (GmbH), Ingolstädter Landstr. 1, 85764 Neuherberg, Germany; 4German Center for Diabetes Research, Ingolstädter Landstr. 1, 85764 Neuherberg, Germany; 5Institute for Biometrics and Epidemiology, German Diabetes Center, Leibniz Center for Diabetes Research at Heinrich Heine University Düsseldorf, Auf’m Hennekamp 65, 40225 Düsseldorf, Germany; 6German Centre for Cardiovascular Research, Partner Site Munich Heart Alliance, Pettenkoferstr. 8a & 9, 80336 Munich, Germany; 7Deutsches Herzzentrum München, Technische Universität München, Lazarettstr. 36, 80636 Munich, Germany; 8Institute of Epidemiology and Medical Biometry, University of Ulm, Helmholtzstr. 22, 89081 Ulm, Germany; 9Else Kröner-Fresenius-Center for Nutritional Medicine, TUM School of Life Sciences, Technical University of Munich, 85354 Freising, Germany; 10Institute of Nutritional Medicine, School of Medicine, Technical University of Munich, Georg-Brauchle-Ring 62, 80992 Munich, Germany; 11Institute for Medical Information Processing, Biometry, and Epidemiology, Ludwig-Maximilians-Univesität München, Marchioninistr. 15, 81377 Munich, Germany

**Keywords:** metabotype, cluster analysis, parameter selection, clinical marker, metabolic diseases, cardiovascular diseases

## Abstract

The aim of metabotyping is to categorize individuals into metabolically similar groups. Earlier studies that explored metabotyping used numerous parameters, which made it less transferable to apply. Therefore, this study aimed to identify metabotypes based on a set of standard laboratory parameters that are regularly determined in clinical practice. K-means cluster analysis was used to group 3001 adults from the KORA F4 cohort into three clusters. We identified the clustering parameters through variable importance methods, without including any specific disease endpoint. Several unique combinations of selected parameters were used to create different metabotype models. Metabotype models were then described and evaluated, based on various metabolic parameters and on the incidence of cardiometabolic diseases. As a result, two optimal models were identified: a model composed of five parameters, which were fasting glucose, HDLc, non-HDLc, uric acid, and BMI (the metabolic disease model) for clustering; and a model that included four parameters, which were fasting glucose, HDLc, non-HDLc, and triglycerides (the cardiovascular disease model). These identified metabotypes are based on a few common parameters that are measured in everyday clinical practice. These metabotypes are cost-effective, and can be easily applied on a large scale in order to identify specific risk groups that can benefit most from measures to prevent cardiometabolic diseases, such as dietary recommendations and lifestyle interventions.

## 1. Introduction

Metabotyping describes the process of forming subgroups based on similarities in subjects’ metabolic or phenotypic characteristics. These subgroups are termed as metabotypes or metabolic phenotypes [1,2,3,4]. All individuals within a subgroup show a high metabolic similarity, while the different subgroups are all distinct from each other. This allows for the identification and description of specific subgroups according to their cardiometabolic disease risk [5,6,7,8]. Evidence suggests that dietary recommendations that are provided at personalized and metabotype levels tend to be more effective than providing general dietary advice [1,9,10,11]. Thus, metabotyping is a promising approach for the development of personalized preventive measures, such as dietary recommendations and lifestyle interventions [1,2,6,12].

Several studies have been performed to define metabotypes [2,3,13]. However, due to the use of different methods and inconsistent definitions, studies have shown large heterogeneities in the types and numbers of parameters used to identify metabotypes or metabolic phenotypes [3]. Some studies have even used a large number of metabolic variables from different metabolic pathways, leading to comprehensive metabotyping [14,15,16,17,18]. Similarly, in our previous study by Riedl and co-authors [15], we identified comprehensive metabotypes in the German population-based KORA study using a range of biochemical and anthropometric parameters.

However, many metabolic parameters are not routinely measured in primary care, making it difficult to implement a comprehensive metabotype concept in general research settings. Therefore, in order to identify a metabotype concept that can be broadly applicable in daily practice, a set of routinely measured clinical parameters, so-called “standard laboratory parameters”, should be explored. The importance of having an easily applicable metabotype definition for use on a large scale to identify subjects with a specific cardiometabolic disease risk, has also been highlighted in a recent perspective paper by Palmnäs et al. [12]. Currently, only a few studies have investigated the use of a reduced set of available parameters to identify metabotypes [5,8,19,20]. In this study, we aimed to develop a statistically guided selection of variables for metabotypes, without disregarding the availability and clinical relevance of parameters.

Therefore, this study aimed to optimize the metabotype definition by reducing the clinical parameters to a few that were economical and routinely measured. For this purpose, we used machine learning-based variable importance methods to assess the suitability of parameters for identifying different metabotypes. In order to evaluate the results, we described the metabotype clusters using various metabolic parameters, as well as the incidences of various cardiometabolic diseases.

## 2. Materials and Methods

### 2.1. Study Population

All data for this study were obtained from the population-based KORA F4 (2006–2008) and KORA FF4 (2013/2014) studies. The data set represents the first and second follow-up examinations of the KORA S4 study, conducted between 1999 and 2001 (n = 4261 participants aged 25–74 years) in the region of Augsburg in southern Germany [21]. In total, 3080 individuals took part in the KORA F4 study, and 2279 individuals participated in the 7-year follow-up examination (KORA FF4). Among these, 2161 individuals participated in both the KORA F4 and KORA FF4 studies. In both studies, participants were invited to the study center where bio-samples were collected, and trained study nurses performed standardized physical examinations as well as computer-assisted face-to-face interviews. Likewise, all participants answered self-administered questionnaires. An in-depth description of the primary study design [21] and of the KORA F4 [22,23] and KORA FF4 [24] studies was reported previously. Written informed consent was provided by all participants, and the studies were approved by the Ethics Committee of the Bavarian Medical Association, and were conducted in accordance with the Declaration of Helsinki.

### 2.2. Biochemical and Anthropometric Parameters

We identified the metabotypes based on fasting biochemical parameters, along with body mass index (BMI) data that were available from the KORA F4 study. BMI was used as a continuous measure in kg/m^2^. Parameters such as high-density lipoprotein cholesterol (HDLc), total cholesterol (TC), triglycerides (TG), glucose, insulin, uric acid, high-sensitive C-reactive protein (hs-CRP), gamma-glutamyltransferase (GGT), glutamate-pyruvate transaminase (GPT), glutamate-oxaloacetate transaminase (GOT), and alkaline phosphatase (AP), were measured in serum samples. Non-HDLc cholesterol was calculated by subtracting HDLc from TC. Leukocyte count and glycated hemoglobin (HbA1c) were measured from fresh venous whole EDTA blood samples. More technical details on the handling of blood samples and the derivation of biomarker measurements can be found elsewhere [15].

### 2.3. Socio-Demographic and Lifestyle Variables

Socio-demographic data included sex, age (in years), and education; the latter was categorized according to the German education system into < 10 years, 10–<12 years, and ≥ 12 years at school. Lifestyle data included physical activity (active: active for ≥2 h per week; inactive: active for <2 h per week), smoking status (smoker, ex-smoker, and never-smoker), and alcohol consumption (≥40 g/day, 20–<40 g/day, 0–<20 g/day, and 0g/day). According to the WHO [25], BMI was categorized into underweight (BMI < 18.5 kg/m^2^), normal weight (BMI 18.5–<25 kg/m^2^), overweight (BMI 25–<30 kg/m^2^), and obese (BMI ≥ 30 kg/m^2^).

### 2.4. Health Status

In both the F4 and FF4 studies, cardiometabolic diseases were assessed during standardized face-to-face computer-assisted interviews and physical examinations. Metabolic diseases were defined as follows: hypertension by a blood pressure of ≥140/90 mmHg in the resting state during physical examination or treatment with antihypertensive medication; type 2 diabetes mellitus was defined by self-reported diagnosis validated by the respective treating physician and by current intake of glucose-lowering medication. In addition, undiagnosed diabetes cases were identified through oral glucose tolerance test based on ADA criteria. Cases with a diagnosis of dyslipidemia or hyperuricemia/gout were defined by the current intake of lipid-lowering drugs or hyperuricemia/gout medication, respectively. Similarly to our previous studies [8,15], we analyzed all metabolic diseases individually, and as a combined outcome variable, “any metabolic diseases” (defined as suffering from at least one of the four metabolic diseases: hypertension, type 2 diabetes, dyslipidemia, and hyperuricemia/gout). Similarly, the diagnosis of myocardial infarction or stroke was defined on the basis of self-reports, and was further validated by means of medical records such as hospital or general practitioners’ records. Both myocardial infarction and stroke were analyzed individually and summarized as a dichotomous variable, “any cardiovascular diseases” (defined as suffering from at least one of two cardiovascular diseases: myocardial infarction and/or stroke).

Prevalent cases were cases identified in the KORA F4 study, and incident cases were defined as newly occurring cases after a follow-up of seven years in the KORA FF4 study (in those participants who were not yet diagnosed with the respective disease in the KORA F4 study).

### 2.5. Data Preprocessing

Among the 3080 participants in the F4 data, we excluded 61 participants, as 54 participants did not fast for at least 8 h before blood collection, and 7 participants had missing information regarding fasting glucose levels. Furthermore, we excluded 18 participants who had more than 10% missing data for the above-mentioned parameters; utimately, there were 3001 study participants in total. Among these, 2120 participated in both the F4 study and the seven-year follow-up FF4 study (Figure 1). We imputed the remaining missing variables of the biomarkers using the multivariate imputation by chained equations ‘mice’ package version 3.8.0 in R [26], which generated five complete data sets with ten iterations each. Subsequently, to avoid the biases of different scales and units, we z-standardized all biochemical and anthropometric parameters before using these imputed and standardized data for clustering purposes only.

### 2.6. Descriptive Statistics

We reported the baseline characteristics, including socio-demographic, lifestyle, and health status, of the KORA F4 study population in total and stratified these by sex. Median and interquartile range (IQR) were shown for continuous variables, and absolute frequency and percentage for categorical variables. In order to analyze the differences in the distributions between sexes as well as metabotypes, we used the Kruskal–Wallis test for continuous variables, and Pearson’s chi-square test for categorical variables. Additionally, we also carried out respective post hoc tests with the Bonferroni correction to examine differences in metabolic parameters between metabotype clusters. As there was missing information for some participants, the maximum number of data available was used, leading to different sample sizes; the exact numbers are provided in the footnotes of the tables.

### 2.7. Parameter Selection

In our previously published studies, we developed the metabotyping concept using a comprehensive set of 32 [15] and 16 [27] widely available clinical parameters. However, in this study, we optimized the metabotype concept by reducing the metabotyping parameters to a few relevant standard clinical parameters. As an initial step, we reduced the 16 parameters to 14 parameters by replacing three-biochemical parameters, TC, low-density lipoprotein (LDL) cholesterol, and the TC/HDL ratio, with non-HDLc cholesterol, as recent findings showed non-HDLc to have high prognostic value [28,29,30,31].

In order to investigate the contribution of individual variables to metabotyping, variable importance analysis was performed using a machine learning-based method, beginning with the 14 parameters. We applied a commonly used feature selection method for biomarker discovery [32,33,34] called permutation variable importance (PVI) [35]. It was implemented using the R function “PIMP” (algorithm for the permutation variable importance measure) [36]. In this method, variable importance is calculated with the help of the variable importance measure of the random forest (RF) algorithm [37]. Initially, an RF model is trained on the original data set. Then, variable importance of each variable is calculated using a decrease in Gini impurity, which is the likelihood of falsely identifying the occurrence of a random variable (for details see [37]). Then, the outcome variable is randomly permuted a fixed number of times. For each permutation of the outcome variable, the variable importance for each predictor variable is calculated, which is referred to as “null importance”. Then, a probability distribution (selected using Kolmogorov–Smirnov tests) is fitted to the null importances. The fitted distribution is then used to derive *p*-values of true importance from the null importances. In this study, we obtained the variable importance by creating 500 trees with all 14 parameters included, and derived the *p*-value of the predictors by randomly permuting the outcome variables 100 times, as described in detail by Altman et al. [35].

As a sensitivity analysis and to validate the results of the PVI method, we applied two other methods to examine variable importance. First, we used the cross-validated permutation variable importance measure (CVPVI), which is an average of all k-fold cross-validated permutation importances [38]. In this method, data sets are divided into k equal folds; for each fold, an RF model is trained. The prediction error from each tree in the RF model is calculated. The same is repeated after permuting each predictor variable. The difference between the two prediction errors is calculated and averaged over all trees. Finally, the differences from all k folds are averaged, and the final relevance of predictor variables is assessed. We implemented this function in R using the “CVPVI” function from the “Vita” package [39]. In our implementation, we carried out 10-fold cross-validation, and created 1000 trees in each fold. The predictor variables were permuted 100 times. For the second method, we performed gradient-boosted feature selection [40]. Gradient boosting is a boosted tree-based supervised learning algorithm [41]. In this method, variable importance is calculated using the fractional contribution each variable provides to the model, based on the total gain of the variable’s splits. These importance scores are then averaged across all decision trees within the model [41,42]. For this task, we used the R package “xgboost” [43].

Based on the top 50% of contributing variables in all methods, as well as their availability in general primary care, we selected a subset of 7 out of the 14 variables.

### 2.8. Metabotyping

Similarly to previous studies [5,15,19,27], we identified metabolically homogenous subgroups (metabotypes) by performing a clustering method called k-means clustering algorithm. Likewise, in order to identify the appropriate number of clusters for the k-means algorithm, we used a statistical function in R called “NbClust” which provides 30 indices for determining the optimal number of clusters [44]. We used the clustering algorithm in all five imputed data sets via the R package “miclust” (multiple imputation in cluster analysis), version 1.2.5 [45].

We used the seven identified parameters (namely TG, BMI, uric acid, fasting glucose, insulin, HDLc, and Non-HDLc) to derive metabotypes for KORA F4 participants. We created several random unique combinations of parameters and computed cluster analyses in each combination, which resulted in different metabotype models. The models with at least 5% or 150 participants in the smallest cluster were regarded as acceptable metabotype models, and were included in this study. In the majority of the models, a three-cluster solution was the best option, followed by two clusters; all other options were ranked low. Similar results were obtained in our previous studies, where three clusters were identified as an appropriate number of clusters [8,15,46]. Therefore, to make the models consistent and comparable, we derived a three-cluster solution in all models.

In each model, we termed cluster 1 as the cluster with the metabolically most favorable clustering biochemical parameters (“healthy metabotype”), in contrast with cluster 3 which was characterized by the metabolically least favorable clustering parameters (“unfavorable metabotype”). Cluster 2 was termed as the intermediate cluster, where the clustering biochemical parameters were in between those of clusters 1 and 3 (“intermediate metabotype”).

The incidence of cardiometabolic disease in KORA FF4 data was used to identify the most appropriate metabotype models that were identified in KORA F4 participants. However, the diseases were not included in the metabotyping process. We ranked the models based on the highest incidence of all metabolic diseases and cardiovascular diseases in cluster 3. The model with the highest rank for “any metabolic disease” was regarded as the best model for metabolic disease, and the model with the highest rank for “any cardiovascular disease” was regarded as the best model for cardiovascular disease. These models were further evaluated on the basis of various metabolic parameters.

We performed all statistical analyses for this study using the statistical software R version 3.6.2 (R Development Core Team, 2010, http://www.r-project.org, (accessed on 17 February 2020) and RStudio Version 1.1.423, which is an integrated development environment (IDE) for R. All tests were two-tailed, and we considered a *p*-value < 0.05 to be statistically significant.

## 3. Results

Table 1 describes the baseline characteristics of the study population, including both demographic parameters and data on the prevalence of diseases identified in the KORA F4 study, in total and stratified by gender. Among the total study population, 52% were female and 48% were male, with a median age of 56 years (IQR = 22 years). The median BMI was 27 kg/m^2^ (IQR = 5.9 kg/m^2^), and almost 55% of the total study population was physically active. The prevalence of “any metabolic disease” and “any cardiovascular disease” was 43.6% and 4.7%, respectively. Compared to men, women had a lower median age and BMI, were more often never-smokers, consumed less alcohol, and showed a lower prevalence of diseases. We observed similar differences between groups in the follow-up study population (KORA FF4) as well (Appendix A).

Figure 2 presents the variable importance of the grouping parameters included in the 14-parameter model. All three methods showed similar results. The seven most important parameters out of fourteen were TG, uric acid, BMI, HDLc, glucose, insulin, and non-HDLc. According to the PVI method, all seven selected parameters had a significant effect. TG was identified as the most important variable in all methods, whereas uric acid was identified as the second most important parameter in two (PVI and CVPVI) methods. Next, BMI, HDLc, and glucose were identified as the third, or either fourth, or fifth most important parameters, respectively, which was followed by insulin. Non-HDLc was also identified as either the sixth or the eighth most important variable. Except for insulin, all identified parameters were labeled a priori as standard laboratory parameters. Therefore, insulin was not included any further in our models. The variable importance carried out on the comprehensive set of 29 parameters also showed similar results (Appendix A). Following parameter selection, we explored unique combinations of the seven selected parameters that resulted in 18 different metabotype models. All models were described on the basis of the cumulative incidence of diseases in the most unfavorable cluster (cluster 3). Compared to the model with 14 parameters, all 18 models based on the selected parameters resulted in comparatively higher incidences of both metabolic and cardiovascular disease in participants of cluster 3 (Appendix A). Among all models, cluster 3 of model 7 showed the highest incidence (62%) of “any metabolic disease”, whereas cluster 3 of model 17 revealed the highest incidence of “any cardiovascular disease” (9.1%) (Appendix A). As a result, these two models were selected for further exploration.

Table 2 shows the distribution of socio-demographic variables across all three clusters for the best models, models 7 and 17. In model 7, about 40% (n = 1189) of participants were assigned to cluster 1, 48% (n = 1140) to cluster 2, and 12% (n = 372) to cluster 3. Meanwhile, in model 17, 42% (n = 1253) were assigned to cluster 1, 49% (n = 1467) to cluster 2, and 9% (n = 281) to cluster 3. In both models, a high proportion of men (60% to 70%) were in cluster 3, whereas a high proportion of women (~70%) were in cluster 1. In both models, cluster 3 had the highest median age (65 years and 64 years, respectively) and median BMI (33.2 and 30.5 kg/m^2^, respectively). Similarly, almost 60% of the participants in cluster 3 were physically inactive. Moreover, participants in cluster 3 were more often heavy drinkers (more than 40 g/day). Compared to cluster 3, cluster 2 and cluster 1 included a higher number of never-smokers and participants with higher education levels.

Table 3 displays the distributions of clinical parameters across the different clusters in the two selected metabotype models. All five clustering parameters in model 7 were significantly different across clusters. Moreover, in both models, other biochemical parameters that were not included in the metabotyping process, such as GGT, GOT, GPT, HbA1c, hs-CRP, AP, insulin, and leukocyte count, also showed significant differences across clusters, with the most unfavorable values in cluster 3.

Table 4 presents the prevalence and incidence of metabolic and cardiovascular diseases of study participants in the three clusters of models 7 and 17, respectively. Regarding the incidence of individual metabolic and cardiovascular diseases in both models, we obtained significant differences across clusters, except for hypertension. Compared to cluster 1 and cluster 2, participants in cluster 3 of both models (models 7 and 17) showed the highest incidence of all metabolic and cardiovascular diseases. This holds for “any metabolic disease” (model 7 (13% vs. 23.7% vs. 62%), model 17 (13.5% vs. 25.3% vs. 58.4%)) and “any cardiovascular disease” (model 7 (1.4% vs. 3.4% vs. 7.4%), model 17 (1.5% vs. 3.3% vs. 9.1%)).

## 4. Discussion

We further improved the metabotyping concept in the population-based KORA F4/FF4 study by using a few routinely measured clinical parameters (namely TG, BMI, uric acid, fasting glucose, HDLc, and non-HDLc) that were identified through the PVI method. By computing a k-means cluster analysis, we identified three-cluster solutions and described them on the basis of metabolic parameters and disease occurrence. We selected two models as the most appropriate solutions.

To the best of our knowledge, this is the first study to assess the importance of parameters for metabotypes, using the PVI method to select them. We further reinforced the results from PVI using two additional variable importance methods (i) CVPVI and (ii) gradient-boosted feature selection. The similar results from these two additional methods validated the results from the PVI method. Furthermore, we created multiple unique subsets of the selected parameters to create 18 different metabotype models, in contrast to using just the initially selected parameters. In accordance with our earlier studies [15,27] and also with other similar papers [5,16,47,48,49,50,51], we used the unsupervised method of k-means clustering for metabotyping, resulting in metabotype models that were independent of disease. Two models out of eighteen were chosen, based on the incidence of disease in the 7-year follow-up KORA FF4 study, and were further analyzed.

Metabotype model 7 was described as the metabotype model with the highest incidence of metabolic diseases in the unfavorable cluster 3, and was based on five parameters (glucose, BMI, uric acid, HDLc, and non-HDLc); meanwhile, cluster 3 of model 17 showed the highest incidence of cardiovascular diseases, and included four parameters (TG, glucose, HDLc, and non-HDLc). Both models 7 and 17 were evaluated using an additional set of biochemical parameters that were not included in identifying the metabotype groups. The concentrations of the biochemical parameters across the three clusters of both models were consistently and significantly different, showing the unique metabolic characteristics of each cluster. This validated our identification of distinct metabotype subgroups (clusters).

The differences regarding the prevalence and incidence of diseases between clusters in both models were statistically significant, except for the incidence of hypertension. Although the incidence of hypertension was higher in cluster 3, it did not reach statistical significance in either model 7 or 17 (*p* = 0.18 and *p* = 0.12, respectively). However, there was a significant difference in the prevalence of hypertension across all three clusters in both models. This may have been due to the high prevalence of hypertension (more than 70%) among participants in cluster 3. Another explanation could also be that there may have been participation bias in the KORA FF4 study, as those who did not participate in the follow-up study were less healthy [52]. Regarding the socio-demographic characteristics, participants in cluster 3 were more likely to have received less than 10 years of education, had a higher median age, a higher BMI, were more physically inactive, and were more often heavy drinkers compared to participants in cluster 2 and cluster 1. Additionally, cluster 3 included the lowest number of current smokers compared to other clusters, but the highest number of ex-smokers and non-drinkers, which likely reflects behavioral changes in response to worsening health status. Thus, these clear differences in risk factors and occurrence of diseases across clusters show that the identified metabotypes represent specific characteristics where the clusters can be meaningfully classified into healthy, intermediate, and unfavorable clusters. Thus, the identified metabotypes can be used as a tool to stratify populations according to their metabolic features. However, the intention of the present research was not to create a risk prediction model; rather, it aimed to define metabolically homogenous subgroups in the population.

We previously used the metabotyping concept to investigate associations between diet and type-2 diabetes (T2D), and identified different associations by metabotype subgroups [27]. T2D risk increased in the healthy subgroup with a higher intake of total meat and processed meat, while in the unhealthy subgroup T2D risk was positively associated with consumption of sugar-sweetened beverages, and inversely associated with fruit intake. We also found significant associations between dietary patterns and T2D in the total population; however, when stratified by the metabotype subgroups, a significant association was only seen in the unhealthy metabotype subgroup [46]. Likewise, in studies by Fiamoncini et al. [53] and O’Sullivan et al. [48], the effect of dietary intervention was only evident after dividing the population into metabolic phenotypes. Similarly, in our recent publication, we successfully applied the metabotypes that were developed in this manuscript in a different study population where participants in different metabotype subgroups showed significantly differential reactions to the oral glucose tolerance test (OGTT) [54]. These studies clearly demonstrate that metabotyping can be used to identify a metabolically similar high-risk subgroup that can benefit from targeted dietary advice and lifestyle intervention.

O’donovan et al. [5] and Hillesheim et al. [19] used decision tree methods in their studies to develop targeted dietary advice for specific metabolic subgroups. When the dietary advice from the decision tree was compared to the individual-based approach that was delivered by a dietitian, they found that the advice matched in more than 80% of the study population. Similar results were seen in the Food4Me study [55], where decision trees were used to provide personalized dietary advice to adults in seven European countries. Providing dietary advice at the individual level is the epitome of personalized nutrition; however, this approach is costly, and involves extensive data collection [19]. On the other had, the metabotype approach provides a simpler and more feasible approach [12]. These results show that using metabotypes can be a promising tool in the field of personalized nutrition. Furthermore, this approach can also help clinicians provide dietary recommendations quickly [5,19] by overcoming the usual barriers such as lack of time, heavy workloads, and inadequate training [56,57].

Several studies have implemented both small and large (n > 50) sets of anthropometric and biochemical parameters, in order to identify metabolically similar groups [3,5,16,17,19,49,58,59,60,61]. Bouwman et al. [14] even included omics data and identified two distinct subgroups to study visualization and identification of health space. As omics data may provide a more comprehensive outlook, we also investigated the inclusion of omics as well as genetics data for metabotyping in one of our previous studies [15]. However, the model did not perform better than the model with the extensive set of 32 biochemical and anthropometric parameters (details not published). A few other studies have also identified metabotypes as a useful measure to identify subjects with a high risk of cardiometabolic disease [48,50,54,62,63,64]. However, these studies did not include a wide range of metabolic and cardiovascular diseases. Furthermore, previous studies were often based on parameters that are not easily accessible in daily practice. For example, in a study by Urpi-Sarda et al. [63], four different subgroups were identified on the basis of a urinary metabolomics fingerprint associated with type-2 diabetes.

The two metabotype models identified in this study are based on a small number of routinely used clinical parameters. We found a distinct difference across the clusters in regard to metabolic parameters. Similarly, the distribution of disease was also different between the subgroups; in addition, the unfavorable cluster 3 showed the highest percentage for both prevalence and incidence of diseases in both models. These consistent results show that our clustering model successfully identified valid metabotypes, which have the advantage of being simpler yet no less valid than previously identified metabotype models [8,15,27]. Moreover, the successful application of the metabotypes identified in this study in a different study population further validates our metabotypes concept [54]. These findings illustrate that identified metabotypes can be easily applied at a population level, as well as in general research settings and primary care, to detect metabolically similar subgroups. Furthermore, they may also help to develop new, targeted, and precise dietary recommendations based on their different metabolic features.

The present study was conducted in a large population-based cohort, which makes the finding of our study generalizable to the adult German population. The use of a few readily available parameters to define valid metabotypes makes the findings of this study simple, cost-effective, and instantly applicable on a large scale, if replicated in other cohorts. We were also able to consider subjects with previously unknown or undiagnosed type 2 diabetes by using an oral glucose tolerance test at baseline and follow-up, which could have otherwise remained undetected. Another strength of this study is that the incidence data on self-reported diseases such as myocardial infarction, stroke, and type 2 diabetes were validated by means of medical records. However, there may have been an underestimation of the prevalence and incidence of dyslipidemia and hyperuricemia, as they were defined only on the bases of reported intake of lipid-lowering drugs and hyperuricemia/gout medication. Additionally, we lost many participants in the follow-up KORA FF4 study, which may have biased our results [52]. The highest dropout rate of participants in cluster 3 may also have underestimated the disease incidence in this cluster.

In conclusion, we successfully identified two valid and practical metabotype solutions based on a minimal number of routinely measured clinical parameters that are generally available in research settings and primary care. Thus, the identified metabotypes can easily be applied to the general population for the purposes of identifying individuals who could benefit from receiving additional preventive measures targeted to metabolic derangements, such as dietary recommendations and lifestyle modifications. The replication of identified metabotypes in a different cohort could further aid in the development of a simple and consistent metabotype definition.

## Figures and Tables

**Figure 1 life-12-01460-f001:**
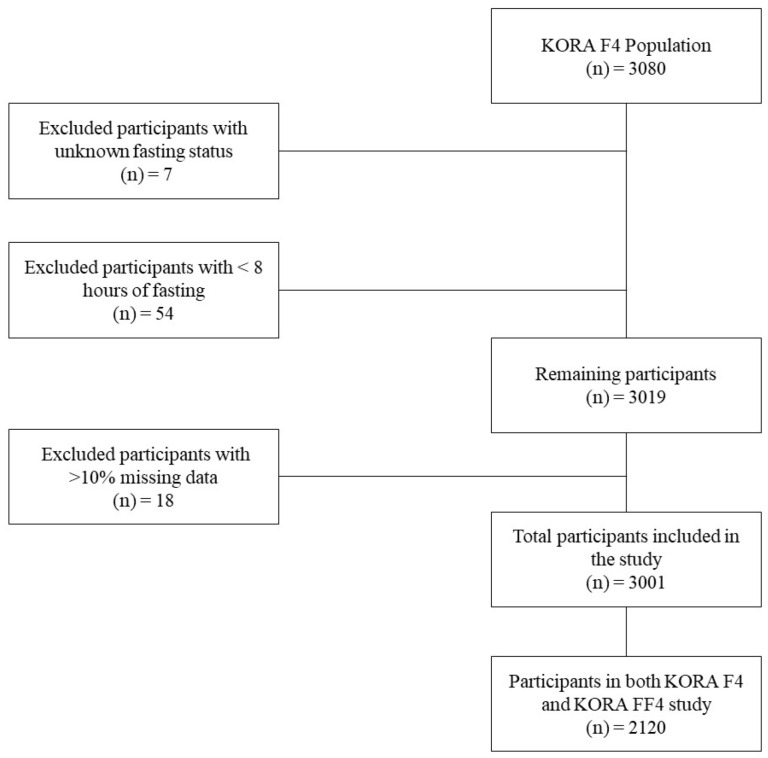
Study flow diagram.

**Figure 2 life-12-01460-f002:**
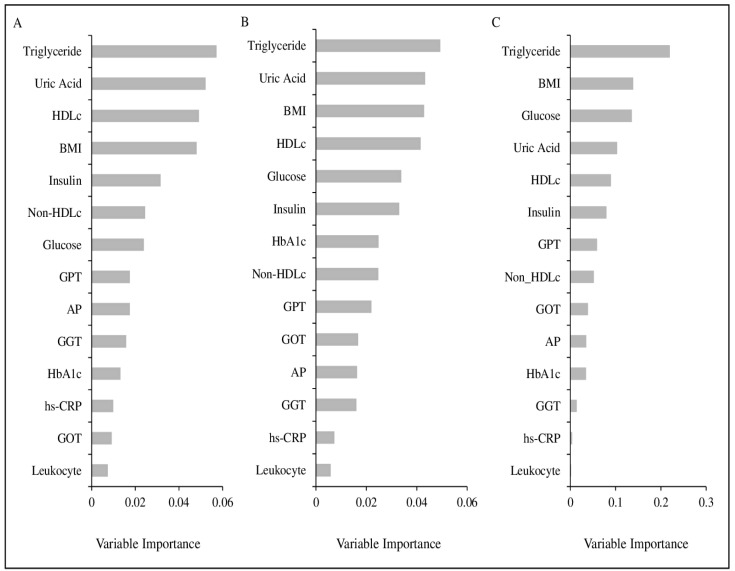
Variable importance of the clustering parameters based on the permutation variable importance (**A**), cross-validated permutation variable importance (**B**), and gradient-boosted feature selection (**C**).

**Table 1 life-12-01460-t001:** Baseline characteristics of the KORA F4 study population.

	Total	Men	Women	***p***-Value
	n = 3001	n = 1450	n = 1551	
**Socio-demographic characteristics**				
Age (years)				0.03
Median (IQR)	56.0 (22.0)	57.0 (23.0)	55.0 (22.0)	
Education				<0.001
<10 years	261 (8.7%)	58 (4.0%)	203 (13.1%)	
10–<12 years	1499 (50.0%)	661 (45.6%)	838 (54.0%)	
≥12 years	1236 (41.2%)	728 (50.2%)	508 (32.8%)	
Missing	5 (0.2%)	3 (0.2%)	2 (0.1%)	
BMI (kg/m^2^)				<0.001
Median (IQR)	27.0 (5.9)	27.3 (5.1)	26.3 (7.1)	
Normal weight (18.5–<25)	941 (31.4%)	345(23.8%)	596 (38.4%)	
Overweight (25–<30)	1253 (41.8%)	726(50.1%)	527 (34.0%)	
Obese (≥30)	793 (26.4%)	375(25.9%)	418 (27.0%)	
Missing	14 (0.5%)	4 (0.3%)	10 (0.6%)	
Physical Activity				0.409
Active	1641 (54.7%)	780 (53.8%)	861 (55.5%)	
Inactive	1356 (45.2%)	666 (45.9%)	690 (44.5%)	
Missing	4 (0.1%)	4 (0.3%)	0 (0.0%)	
Smoking				<0.001
Smoker	524 (17.5%)	281 (19.4%)	243 (15.7%)	
Ex-Smoker	1218 (40.6%)	715 (49.3%)	503 (32.4%)	
Never-Smoker	1254 (41.8%)	450 (31.0%)	804 (51.8%)	
Missing	5 (0.2%)	4 (0.3%)	1 (0.1%)	
Alcohol consumption				<0.001
≥40 g/day	336 (11.2%)	290 (20.0%)	46 (3.0%)	
20–<40 g/day	540 (18.0%)	351 (24.2%)	189 (12.2%)	
0–<20 g/day	1221 (40.7%)	508 (35.0%)	713 (46.0%)	
0 g/day	900 (30.0%)	297 (20.5%)	603 (38.9%)	
Missing	4 (0.1%)	4 (0.3%)	0 (0.0%)	
**Prevalence of disease n (%)**				
Type 2 diabetes mellitus	242 (8.1%)	144 (9.9%)	98 (6.3%)	<0.001
Hypertension	1150 (38.3%)	639 (44.1%)	511 (32.9%)	<0.001
Missing	7 (0.2%)	4 (0.3%)	3 (0.2%)	
Hyperuricemia	113 (3.8%)	90 (6.2%)	23 (1.5%)	<0.001
Missing	2 (0.1%)	2 (0.1%)	0 (0.0%)	
Dyslipidemia	386 (12.9%)	219 (15.1%)	167 (10.8%)	<0.001
Missing	3 (0.1%)	3 (0.2%)	0 (0.0%)	
Any metabolic disease	1309 (43.6%)	730 (50.3%)	579 (37.3%)	<0.001
Missing	8 (0.3%)	4 (0.3%)	3 (0.2%)	
Stroke	71 (2.4%)	46 (3.2%)	25 (1.6%)	0.007
Myocardial infarction	79 (2.6%)	62 (4.3%)	17 (1.1%)	<0.001
Any cardiovascular disease	142 (4.7%)	100 (6.9%)	42 (2.7%)	<0.001

Median (IQR) for continuous variables and n (column %) for categorical variables. *p*-values are from the Kruskal–Wallis test for continuous variables, and from Pearson’s chi-squared test for categorical variables. Prevalence: due to missing information, there were reduced data sets for hypertension n = 2994, hyperuricemia n = 2999, dyslipidemia n = 2998, and “any metabolic disease” n = 2993. KORA, Cooperative Health Research in the Region of Augsburg.

**Table 2 life-12-01460-t002:** Characteristics of the study population across the three clusters in model 7 and model 17, KORA F4 study.

	Total		Metabotype		***p***-Value
		Cluster 1	Cluster 2	Cluster 3	
**Model 7**	n = 3001	n = 1189	n = 1440	n = 372	
Age	56.0 (22.0)	51.0 (20.0)	57.0 (22.0)	**65.0 (15.0)**	<0.001
Median (IQR)					
Sex					<0.001
Male	1450 (48.3%)	282 (23.7%)	**942 (65.4%)**	226 (60.8%)	
Female	1551 (51.7%)	**907 (76.3%)**	498 (34.6%)	146 (39.2%)	
Education					<0.001
<10 years	261 (8.7%)	86 (7.2%)	124 (8.6%)	**51 (13.7%)**	
10–<12 years	1499 (50.0%)	584 (49.2%)	721 (50.1%)	**194 (52.2%)**	
≥12 years	1236 (41.3%)	**516 (43.5%)**	593 (41.2%)	127 (34.1%)	
BMI					<0.001 <0.001
Median (IQR)	27.0 (5.9)	24.2 (3.9)	28.2 (4.6)	**33.2 (6.5)**	
Normal weight (18.5–<25)	941 (31.5%)	**723 (61.2%)**	205 (14.3%)	13 (3.5%)	
Overweight (25–<30)	1253 (41.9%)	387 (32.8%)	**793 (55.2%)**	73 (19.8%)	
Obese (≥30)	793 (26.5%)	71 (6.0%)	439 (30.5%)	**283 (76.7%)**	
Physical Activity					<0.001
Active	1641 (54.8%)	**719 (60.5%)**	767 (53.4%)	155 (41.7%)	
Inactive	1356 (45.2%)	469 (39.5%)	670 (46.6%)	**217 (58.3%)**	
Smoking					<0.001
Smoker	524 (17.5%)	210 (17.7%)	**282 (19.6%)**	32 (8.6%)	
Ex-smoker	1218 (40.7%)	419 (35.3%)	594 (41.3%)	**205 (55.1%)**	
Never smoker	1254 (41.9%)	**558 (47.0%)**	561 (39.0%)	135 (36.3%)	
Alcohol consumption					<0.001
≥40 g/day	336 (11.2%)	97 (8.2%)	181 (12.6%)	**58 (15.6%)**	
20–<40 g/day	540 (18.0%)	211 (17.8%)	**273 (19.0%)**	56 (15.1%)	
0–<20 g/day	1221 (40.7%)	**534 (44.9%)**	561 (39.0%)	126 (33.9%)	
0 g/day	900 (30.0%)	346 (29.1%)	422 (29.4%)	**132 (35.5%)**	
**Model 17**	n = 3001	n = 1253	n = 1476	n = 281	
Age					<0.001
Median (IQR)	56.0 (22.0)	52.0 (22.0)	57.0 (21.0)	**64.0 (16.0)**	
Sex					<0.001
Male	1450 (48.3%)	384 (30.6%)	868 (59.2%)	**198 (70.5%)**	
Female	1551 (51.7%)	**869 (69.4%)**	599 (40.8%)	83 (29.5%)	
Education					0.024
<10 years	261 (8.7%)	99 (7.9%)	132 (9.0%)	**30 (10.7%)**	
10–<12 years	1499 (50.0%)	620 (49.6%)	719 (49.1%)	**160 (56.9%)**	
≥12 years	1236 (41.3%)	**532 (42.5%)**	613 (41.9%)	91 (32.4%)	
BMI					<0.001 <0.001
Median (IQR)	27.0 (5.9)	25.0 (5.3)	28.0 (5.2)	**30.5 (5.9)**	
Normal weight (18.5–<25)	941 (31.5%)	**632 (50.6%)**	287 (19.7%)	22 (7.9%)	
Overweight (25–<30)	1253 (41.9%)	437 (35.0%)	**716 (49.0%)**	100 (36.0%)	
Obese (≥30)	793 (26.5%)	180 (14.4%)	457 (31.3%)	**156 (56.1%)**	
Physical Activity					<0.001
Active	1641 (54.8%)	**763 (61.0%)**	763 (52.1%)	115 (40.9%)	
Inactive	1356 (45.2%)	488 (39.0%)	702 (47.9%)	**166 (59.1%)**	
Smoking					<0.001
Smoker	524 (17.5%)	195 (15.6%)	**277 (18.9%)**	52 (18.5%)	
Ex-smoker	1218 (40.7%)	476 (38.1%)	604 (41.2%)	**138 (49.1%)**	
Never smoker	1254 (41.9%)	**579 (46.3%)**	584 (39.9%)	91 (32.4%)	
Alcohol consumption					<0.001
≥40 g/day	336 (11.2%)	128 (10.2%)	155 (10.6%)	**53 (18.9%)**	
20–<40 g/day	540 (18.0%)	212 (16.9%)	**282 (19.2%)**	46 (16.4%)	
0–<20 g/day	1221 (40.7%)	**568 (45.4%)**	560 (38.2%)	93 (33.1%)	
0 g/day	900 (30.0%)	343 (27.4%)	**468 (31.9%)**	89 (31.7%)	

Median (IQR) for continuous variables and n (column %) for categorical variables, NA excluded. *p*-values are from the Kruskal–Wallis test for continuous variables and from Pearson’s chi-squared test for categorical variables. Model 7 included 5 parameters (glucose, BMI, uric acid, HDLc, and non-HDLc), and Model 17 included 4 parameters (glucose, triglyceride, HDLc, and non-HDLc), and BMI body mass index. Due to missing information, there were reduced data sets for education n = 2996, BMI n = 2987, physical activity n = 2997, smoking n = 2996, and alcohol consumption n = 2997. The highest values across the clusters are marked in bold. KORA, Cooperative Health Research in the Region of Augsburg. KORA, Cooperative Health Research in the Region of Augsburg.

**Table 3 life-12-01460-t003:** Comparison of the clustering parameters and other metabolic parameters across the three clusters of two selected clustering models (7 and 17), KORA F4 study.

	Total	Metabotype			***p***-Value
		Cluster 1	Cluster 2	Cluster 3	
**Model 7**	n = 3001	n = 1189	n = 1440	n = 372	
Parameters used for metabotyping					
BMI [kg/m^2^]	26.90 (5.93)	24.19 (4.31) ^a^	28.24 (4.85) ^b^	**33.17 (6.47) ^c^**	<0.001
Uric acid [µmol/L]	299.41 (114.12)	243.52 (79.29) ^a^	334.12 (93.88) ^b^	**375.29 (113.76) ^c^**	<0.001
Glucose [mg/dL]	94.00 (14.00)	89.00 (10.80) ^a^	96.00 (12.40) ^b^	**122.00 (28.40) ^c^**	<0.001
HDLc [mmol/L]	1.39 (0.52)	**1.70 (0.45) ^a^**	1.26 (0.37) ^b^	1.18 (0.39) ^c^	<0.001
Non-HDLc [mmol/L]	4.05 (1.32)	3.59 (1.21) ^a^	**4.49 (1.28) ^b^**	4.01(1.25) ^c^	<0.001
Other Parameters					
TG [mmol/L]	1.19 (0.90)	0.82 (0.56) ^a^	1.46 (0.87) ^b^	**1.76 (1.22) ^c^**	<0.001
AP [µmol/L]	1.10 (0.42)	1.00 (0.43) ^a^	1.14 (0.39) ^b^	**1.21 (0.43) ^c^**	<0.001
GPT [µkat/L]	0.35 (0.23)	0.28 (0.16) ^a^	0.40 (0.24) ^b^	**0.48 (0.32) ^c^**	<0.001
GOT [µkat/L]	0.41 (0.14)	0.38 (0.12) ^a^	0.43 (0.14) ^b^	**0.45 (0.19) ^c^**	<0.001
GGT [µkat/L]	0.43 (0.4)	0.31 (0.26) ^a^	0.50 (0.42) ^b^	**0.63 (0.53) ^c^**	<0.001
HbA1c [%]	5.50 (0.5)	5.30 (0.42) ^a^	5.50 (0.5) ^b^	**6.10 (0.98) ^c^**	<0.001
hs-CRP [mg/L]	1.18 (2.03)	0.76 (1.38) ^a^	1.38 (2.08) ^b^	**2.49 (3.27) ^c^**	<0.001
Leukocytes (n/L)	5.70 (2)	5.30 (1.84) ^a^	5.90 (1.93) ^b^	**6.30 (2) ^c^**	<0.001
Insulin [µU/mL]	8.80 (6.70)	6.60 (4.14) ^a^	10.00 (6.46) ^b^	**18.00 (11.74) ^c^**	<0.001
**Model 17**	n = 3001	n = 1253	n = 1467	n = 281	
Parameters used for metabotyping					
TG [mmol/L]	1.19 (0.90)	1.18 (0.90) ^a^	1.47 (0.78) ^b^	**2.71 (1.74) ^c^**	<0.001
Glucose [mg/dL]	94.00 (14.00)	90.00 (11.60) ^a^	96.00 (13.20) ^b^	**124.00 (38.00) ^c^**	<0.001
HDLc [mmol/L]	1.39 (0.52)	**1.70 (0.45) ^a^**	1.24 (0.37) ^b^	1.08 (0.35) ^c^	<0.001
Non-HDLc [mmol/L]	4.05 (1.32)	3.48 (1.05) ^a^	4.49 (1.19) ^b^	**4.49 (1.27) ^b^**	<0.001
Other Parameters					
BMI [kg/m^2^]	26.99 (5.93)	24.95 (5.42) ^a^	27.95 (5.35) ^b^	**30.54 (5.93) ^c^**	<0.001
Uric acid [µmol/L]	299.41 (114.12)	260.5 (96.70) ^a^	321.17 (107.05) ^b^	**378.23 (117.8) ^c^**	<0.001
AP [µmol/L]	1.10 (0.42)	1.01 (0.42) ^a^	1.15 (0.41) ^b^	**1.20 (0.44)^b^**	<0.001
GPT [µkat/L]	0.35 (0.23)	0.30 (0.18) ^a^	0.39 (0.24) ^b^	**0.48 (0.31) ^c^**	<0.001
GOT [µkat/L]	0.41 (0.14)	0.39 (0.13) ^a^	0.42 (0.15) ^b^	**0.45 (0.19) ^c^**	<0.001
GGT [µkat/L]	0.43 (0.40)	0.34 (0.28) ^a^	0.48 (0.42) ^b^	**0.71 (0.59) ^c^**	<0.001
HbA1c [%]	5.50 (0.50)	5.40 (0.50) ^a^	5.50 (0.50) ^b^	**6.20 (1.22) ^c^**	<0.001
hs-CRP [mg/L]	1.18 (2.03)	0.86 (1.55) ^a^	1.39 (2.23) ^b^	**1.97 (2.57)**	<0.001
Leukocytes (n/L)	5.70 (2.00)	5.40 (1.84) ^a^	5.90 (2.00) ^b^	**6.40 (2.16)**	<0.001
Insulin [µU/mL]	8.80 (6.70)	6.90 (4.62) ^a^	10.00 (6.54) ^b^	**17.00 (10.94)**	<0.001

Median and interquartile range (IQR) were calculated through the mean of median and IQR in all five imputed data sets. *p*-values are from the Kruskal–Wallis test. Different superscript letters represent a significant difference between clusters obtained from the Kruskal–Wallis post hoc test with Bonferroni correction. The highest median values across the three clusters are marked in bold. BMI: body mass index; HDLc: high-density lipoprotein; TG: triglyceride; GGT: gamma-glutamyltransferase; GOT: glutamate-oxaloacetate transaminase; GPT: glutamate-pyruvate transaminase; HbA1c: glycated hemoglobin, hs-CRP: high-sensitive C-reactive protein, AP; alkaline phosphatase. KORA, Cooperative Health Research in the Region of Augsburg.

**Table 4 life-12-01460-t004:** Prevalence and incidence of diseases across the three clusters of two selected clustering models (7 and 17), KORA F4 and FF4 studies.

	Total		Metabotype		***p***-Value
		Cluster 1	Cluster 2	Cluster 3	
**Model 7**	n = 3001	n = 1189	n = 1440	n = 372	
Prevalence of disease in KORA F4; n (%)					
Type 2 diabetes	242 (8.06%)	15 (1.26%)	52 (3.6%)	**175 (47.0%)**	<0.001
Hypertension	1150 (38.4%)	249 (21.0%)	616 (42.9%)	**285 (76.6%)**	<0.001
Hyperuricemia	113 (3.8%)	15 (1.3%)	58 (4.0%)	**40 (10.8%)**	<0.001
Dyslipidemia	386 (12.9%)	101 (8.5%)	173 (12.0%)	**112 (30.2%)**	<0.001
Any metabolic diseases	1309 (43.7%)	299 (25.2%)	684 (47.6%)	**326 (87.9%)**	<0.001
Stroke	71 (2.4%)	19 (1.6%)	32 (2.2%)	**20 (5.4%)**	<0.001
Myocardial infraction	79 (2.6%)	12 (1.0%)	38 (2.6%)	**29 (7.8%)**	<0.001
Any cardiovascular disease	142 (4.7%)	27 (2.3%)	68 (4.7%)	**47 (12.6%)**	<0.001
Incidence of disease in KORA FF4; n (%)	n = 2120	n = 895	n = 1003	n = 222	
Type 2 diabetes	94 (4.7%)	13 (1.5%)	43 (4.4%)	**38 (30.2%)**	<0.001
Hypertension	230 (10.9%)	86 (9.6%)	114 (11.4%)	**30 (13.6%)**	0.187
Hyperuricemia	45 (2.1%)	0 (0.0%)	24 (2.4%)	**21 (9.5%)**	<0.001
Dyslipidemia	157 (7.4%)	27 (3.0%)	92 (9.2%)	**38 (17.2%)**	<0.001
Any metabolic diseases	442 (21.9%)	114 (13.0%)	233 (23.7%)	**95 (62.0%)**	<0.001
Stroke	35 (1.7%)	10 (1.1%)	15 (1.5%)	**10 (4.6%)**	0.001
Myocardial infraction	27 (1.3%)	2 (0.2%)	20 (2.0%)	**5 (2.4%)**	<0.001
Any cardiovascular disease	60 (2.9%)	12 (1.4%)	33 (3.4%)	**15 (7.4%)**	<0.001
**Model 17**	n = 3001	n = 1235	n = 1467	n = 281	
Prevalence of disease in KORA F4; n (%)					
Type 2 diabetes	242 (8.1%)	39 (3.1%)	67 (4.6%)	**136 (48.4%)**	<0.001
Hypertension	1150 (38.4%)	332 (26.6%)	620 (42.4%)	**198 (70.5%)**	<0.001
Hyperuricemia	113 (3.8%)	22 (1.8%)	53 (3.6%)	**38 (13.6%)**	<0.001
Dyslipidemia	386 (12.9%)	140 (11.2%)	168 (11.5%)	**78 (27.9%)**	<0.001
Any metabolic diseases	1309 (43.7%)	385 (31.0%)	694 (47.4%)	**230 (82.1%)**	<0.001
Stroke	71 (2.4%)	26 (2.1%)	35 (2.4%)	**10 (3.6%)**	0.334
Myocardial infraction	79 (2.6%)	24 (1.9%)	37 (2.5%)	**18 (6.4%)**	<0.001
Any cardiovascular disease	142 (4.7%)	46 (3.6%)	70 (4.8%)	**26 (9.2%)**	<0.001
Incidence of disease in KORA FF4; n (%)	n = 2120	n = 916	n = 1040	n = 164	
Type 2 diabetes	94 (4.7%)	15 (1.7%)	54 (5.4%)	**25 (26.9%)**	<0.001
Hypertension	230 (10.9%)	85 (9.3%)	125 (12.0%)	**20 (12.3%)**	0.127
Hyperuricemia	45 (2.1%)	9 (0.1%)	29 (2.8%)	**7 (4.3%)**	0.002
Dyslipidemia	157 (7.4%)	30 (3.3%)	92 (8.8%)	**35 (21.5%)**	<0.001
Any metabolic diseases	442 (21.9%)	120 (13.5%)	256 (25.3%)	**66 (58.4%)**	<0.001
Stroke	35 (1.7%)	11 (1.2%)	14 (1.4%)	**10 (6.2%)**	<0.001
Myocardial infraction	27 (1.3%)	2 (0.2%)	21 (2.0%)	**4 (2.6%)**	<0.001
Any cardiovascular disease	60 (2.9%)	13 (1.5%)	33 (3.3%)	**14 (9.1%)**	<0.001

n (column%), NA excluded. *p*-values from Pearson’s chi-squared test (Fisher’s exact test for low frequencies). Model 7 included 5 parameters (glucose, BMI, uric acid, HDLc, and non-HDLc), and Model 17 included 4 parameters (glucose, triglyceride, HDLc, and non-HDLc). Prevalence: due to missing information, there were reduced data sets for hypertension n = 2994, hyperuricemia n = 2999, dyslipidemia n = 2998, and “any metabolic disease” n = 2993. Incidence: due to missing information, there were reduced data sets for type 2 diabetes n = 1988, hypertension n = 2115, hyperuricemia n = 2117, dyslipidemia n = 2117, “any metabolic disease” n = 2017, stroke n = 2091, myocardial infraction n = 2076, and “any cardiovascular disease” n = 2055. The highest prevalence and incidence of diseases across three different clusters are marked in bold. KORA, Cooperative Health Research in the Region of Augsburg.

## Data Availability

The authors confirm that, for approved reasons, access restrictions apply to the data underlying the findings, and thus the information cannot be made freely available in the manuscript, in the Appendix A, or at a public repository. The data are subject to national data protection laws and restrictions that were imposed by the ethics committee of the Bavarian Medical Association (“Bayerische Landesärztekammer”), in order to ensure the data privacy of the study participants, since they did not explicitly consent to the data being made publicly available. Data can be applied through an individual project agreement with KORA that allows researchers to access the data in the same way used by this study’s authors to access the data. Applications for access to the data sets can be made via the KORA-passt platform (https://helmholtz-muenchen.managed-otrs.com/external/, accessed on 10 August 2022).

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
