# Peer review of "Optimized Metabotype Definition Based on a Limited Number of Standard Clinical Parameters in the Population-Based KORA Study"

_life, 2022, doi:10.3390/life12101460_

Round 1

Reviewer 1 Report

In general it is a very interesting piece with relevant results .    The study has a large-enough number of patients to perform such analysis.   Metabotypes identified are convincing according to the statistics and the size of the population. It would be great to see a replication or external validation of them.    Comprehensive literature review and clerly written.    However, I wonder if 3 clustesrs may be limiting, and if perhaps 4 or 5 could even better represent the complexity of the dtaset. I would have liked to see more of this. Or some soft clustering where we can see that some patients are less like other members of the assigned clusters, while others are “typical”.   Some degree or measure of how homogeneous each cluste ris would be a usefull information to have, so that it is possible to assign some degree of ocnfidence to a metabotype over another one.    Perhaps hypertension did reach significance in some of the not-selected models.     Suggested explanations of why hypertension did not reach significance in the selected models are pertinent.  Just a few suggestions:   Line 30: correlated with Line 33: use of the metabotyping concept in general, or use of metabotyping as a concept in… Line 62: described according to their Line 85: statistically-guided selection Line 172: differences in Line 186: method, described in the following.  

Reviewer 2 Report

The authors present a reasonable study to optimize the metabotype definition using a large scale population-based clinical study. The authors performed rigorous statistical analysis and machine learning modeling to investigate the best features for the clinical data. However, there are some major concerns to be addressed. 

1.     The article aims to find an optimal definition of metabotype to group similar metabolic characteristic people into subgroups. The authors used K-means clustering. Is K-means clustering is the golden-standard way to perform the metabotyping? How to determine the best K? How about other clustering methods such as PCA?  The authors should compare the performance across different methods.

2.     The authors performed K-means clustering after feature selection by xgboosting. The features were selected based on the prediction performance by xgboosting. Therefore, the features in the clusters were affected by the prediction modeling performance, which is slightly biased. Could the authors provide the clustering comparison before prediction modeling. In the clinical setting, unsupervised clustering usually performs first instead of supervised modeling. 

Reviewer 3 Report

The topic is very interesting and the acquired results are able to support the conclusions.

The work is clearly presented in a logical manner.

I would advise more validation steps to be implemented, to efface any bias concerns, such as roc curves.

Finally the authors could also discuss the utility of using more data such as metabolomic, and what this could allow further to elaborate, if important correlations were pinpointed.

I strongly advise the publication of this manuscript after the aforementioned comments.

Round 2

Reviewer 2 Report

The authors provide sufficient evidence to address my comments. I recommend accepting the manuscript after text editing.